# Improving visualization of the cervix during pelvic exams: A simulation using a physical model of the speculum and human vagina as a steppingstone to reducing disparities in gynecological cancers

**Rahul Sai Yerrabelli**[1,2]*, **Peggy K. Palsgaard**[1,2], **Ashkhan Hojati**[1,2], **Amy J. Wagoner Johnson**[1,3,4]

**1** Carle Illinois College of Medicine, University of Illinois at Urbana-Champaign, Champaign, Illinois, United States of America, **2** Carle Foundation Hospital, Urbana, Illinois, United States of America, **3** Grainger College of Engineering, University of Illinois at Urbana Champaign, Urbana, Illinois, United States of America, **4** Carl R. Woese Institute for Genomic Biology, University of Illinois at Urbana Champaign, Urbana, Illinois, United States of America

* rsy2@illinois.edu

**Data Availability Statement:** All data and code files are available from the Zenodo database (Dataset

## Abstract

Pelvic exams are frequently complicated by collapse of the lateral vaginal walls, obstructing the view of the cervix. To overcome this, physicians frequently repurpose a glove or a condom as a sheath placed over the speculum blades to retract the lateral vaginal walls. Despite their regular use in clinical practice, little research has been done comparing the relative efficacy of these methods. Better visualization of the cervix can benefit patients by decreasing examination-related discomfort, improving cancer screening accuracy, and preventing the need to move the examination to the operating room under general anesthesia. This study presents a physical model that simulates vaginal pressure being exerted around a speculum. Using it, we conduct controlled experiments comparing the efficacy of different condom types, glove materials, glove sizes, and techniques to place gloves on the speculum. The results show that the best sheath is the middle finger of nitrile-material gloves. They provide adequate lateral wall retraction without significantly restricting the opening of the speculum. In comparison, condoms provide a smaller amount of retraction due to loosely fitting the speculum. They may still be a reasonable option for a different speculum size. However, vinyl-material gloves are an impractical option for sheaths; they greatly restrict speculum opening, occasionally even breaking the speculum, which overcome its retraction benefits. Glove size, condom brand, and condom material (latex vs polyisoprene) had minimal impact. This study serves as a guide for clinicians as they use easily accessible tools to perform difficult pelvic exams. We recommend that physicians consider nitrile gloves as the preferred option for a sheath around a speculum. Additionally, this study demonstrates proof-of-concept of a physical model that quantitatively describes different materials on their ability to improve cervical access. This model can be used in future research with

doi:10.5281/zenodo.6790210, Code doi:10.5281/zenodo.6790202)

**Funding:** The authors received no specific funding for this work.

**Competing interests:** The authors have declared that no competing interests exist.

more speculum and material combinations, including with materials custom-designed for vaginal retraction.

## Introduction

Pelvic exams are a cornerstone of gynecological care, used in cervical cancer screening [1, 2] as well as for the diagnosis and treatment of a wide variety of conditions that affect millions of patients every year [3]. For many reasons, the pelvic exam can be difficult for both physicians and patients [4]. One challenge gynecologists have struggled with since the beginning of the field [5] is the inward collapse of the lateral vaginal walls during the exam. This can obstruct the physician's view of the cervix making a proper physical exam difficult.

Recall the standard speculum is composed of a handle and two blades—an upper blade and a lower blade (**Fig 1A**). When opened, the blades separate to retract the anterior and posterior vaginal walls respectively [6]. However, nothing provides retraction for the lateral vaginal walls and prevents them from collapsing (bulging) inward. While is acceptable for many patients, the lateral vaginal walls of some patients collapse significantly enough to completely obscure the view (**Fig 1B–1D**). Difficulty in achieving cervical access leads to more maneuvering by the physician, subsequent patient discomfort, potentially incorrect diagnostic results and missed cancers, and occasionally the need to move the exam to the operating room and using general anesthesia. This is especially concerning given the obesity epidemic; 89% of clinicians reported lateral wall collapse as a major reason why cervical sampling is more difficult in patients with obesity [7]. One large study estimated 20% of the cervical cancers in obese and overweight weight undergoing screening could be prevented if cervical visualization in these patients could be improved, which would lead to more adequate tissue sampling [8, 9].

Lateral vaginal wall collapse is an issue well-known to those frequently performing gynecologic care and is often presented in educational materials teaching new trainees how to perform exams [4, 10–19]. However, the peer-reviewed, published literature characterizing the issue is limited. The available articles focus on sophisticated add-ons [20–23], novel speculum designs [5, 12], or avoiding the issue entirely by circumventing the physical exam (favoring urinary [24, 25] or vaginal self-sampling [26–30] instead) or using endoscopic visualization

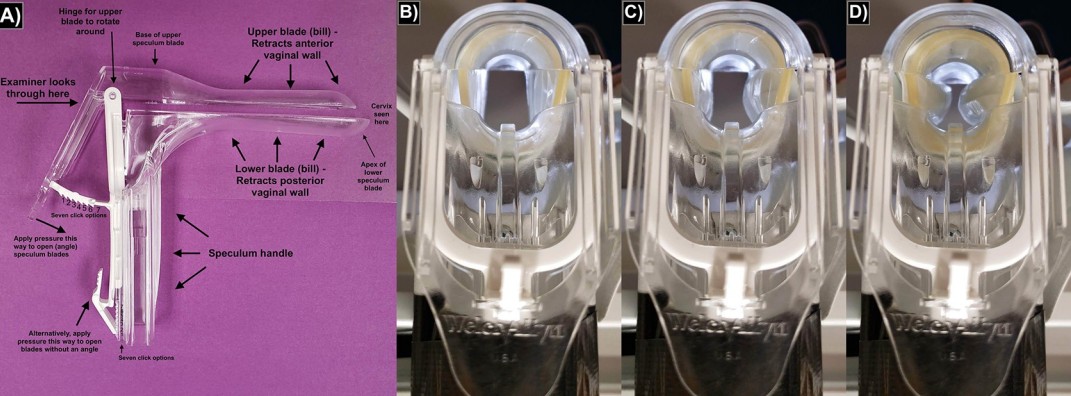

**Fig 1. Speculum photos.** (a) labeled diagram of the Welch Allyn plastic speculum used in our experiments; (b,c,d) the physician's view through the speculum at varying vaginal pressures as represented by our physical model during an experimental trial with a Skyn condom: (b) 0 mmHg with excellent visualization, (c) 40 mmHg (5.3 kPa) with adequate visualization, and (d) 120 (16.0 kPa) mmHg with poor visualization due to lateral wall collapse.

instead of a speculum [21, 31–33]. Despite, or perhaps because of their novelty, these methods are far from becoming the status quo for the standard office gynecologic exam. In practice, the most commonly used method to address collapse of the vaginal walls in the clinic is repurposing supplies easily found in a clinical setting, such as gloves or condoms, to act as a sheath (covering) and surround the blades of speculum. A survey of the members of American Society of Colposcopy and Cervical Pathology showed that 73% of clinicians use this technique, which is more common than sidewall retractors (59%), tenacula (72%), and patient positioning (62%) [7]. Despite their frequent use, to the authors' knowledge, there are only two peer-reviewed article [34, 35] published on PubMed thus far describing how to use these ad hoc solutions and none that rigorously compare the methods, though it is mentioned in passing by some other articles [4, 7, 10–12] and books [13–15].

This study aims to identify the best method for improving visualization of the cervix by decreasing lateral wall collapse using simple tools readily available in a typical clinical setting. These data serve as a guide for practitioners as they navigate ad hoc solutions that are commonly and informally passed along in clinical practice.

## Methods

The specific tools we considered were condoms and gloves. We tested four common condom brands, several glove sizes and material types, and multiple methods of placing the glove on the speculum. These tools were tested with a simulation of an exam using a physical model. This simulation was constructed to mimic the pressure exerted on a speculum by the vaginal walls during a pelvic exam. We also measured the effect that the materials had on restricting opening of the speculum as, if this effect was significant enough, the sheath can paradoxically worsen cervical visualization. As we simulated the exam using a physical model, no human subjects or animals were used in our experiments and specific ethical approval was not needed or sought.

### Materials

We used standard bivalve, clear plastic (acrylic) speculums (KleenSpec® Vaginal Specula, Item No. DYND70401L, Welch Allyn, Inc, UPC #732094143690) designed for outpatient gynecology office practice. These are disposable self-retaining speculums that are manufactured clean, but non-sterile. We tested the use of gloves, condoms, or neither.

For the trials using a glove as the sheath, we tested disposable, non-sterile, nonlatex, clinical-grade gloves. We compared two glove materials: vinyl (Primacare Medical Supplies, UPC #189365002574) and nitrile (STRONG Manufacturers, Black Nitrile Pro Gloves, SMP-75042, SMP-75043, and SMP-75044), the glove types most found in our clinical environment. For the nitrile gloves, we also compared three sizes (small, medium, and large) and compared different techniques to place the glove onto a speculum: both blades into the middle finger, each blade into a separate finger (3rd and 4th fingers), and the blades into the glove's palm with the fingers removed.

In the trials using a condom as a sheath, we tested condoms commercially available for sexual intercourse. These tested four common brands and two condom materials: Trojan (latex), LifeStyles (latex), Durex (synthetic polyisoprene), and Skyn (synthetic polyisoprene).

In addition to the condom and glove sheath trials, a set of trials was also performed without either a glove or condom (no sheath applied) and used as a control.

For each set of experimental conditions, we performed a total of three trials. A new condom or glove was used for every trial. The condoms remained in the packaging until immediately before use to control for any changes to the material or the lubricant drying out.

Please see the supplemental S1 Table for a detailed list of all materials, equipment, and software used.

## Experimental set-up

**Placement of the sheath (for condom sheaths only).** For the condom sheath trials, the condom being tested was put on the blades of a closed speculum and pulled back to where the bases of the speculum blades meet the speculum handle (**Fig 1A**). The closed end of the condom was cut off with scissors to create an opening for visualization. Care was taken to make sure that the created opening did not go past the apex of the upper blade so as to prevent the condom from rolling back towards the speculum base if the speculum was inserted into a vagina.

**Placement of the sheath (for glove sheaths only).** Similar to the condom sheath trials, our glove sheath trials were initiated with a glove placed on a closed speculum; However, there are multiple ways the glove could be placed. For the majority of our trials, we inserted the closed speculum blades specifically into the middle finger of the glove (middle finger method). Then, the speculum blades were slightly opened, and the end of the middle finger was cut off with scissors to allow for visualization between the blades. Alternatively, we also performed trials where the blades were inserted into two adjacent glove fingers, one for each speculum blade (two finger method). For this application technique, we slightly opened the speculum after inserting it into the palm of the glove and then slid the upper (anterior) blade into the third (middle) finger and the lower (posterior) blade in the fourth (ring) finger. We then cut away the glove material between the two fingers for visualization. Thirdly, we tested an application method in which only the palm of the glove was used as the speculum sheath (palm method). However, we abandoned data collection for this midway through testing when it became clear this approach was far inferior to the other two techniques; it provided no lateral retraction and would frequently slip off the speculum blades.

**Setting up the speculum.** To open the speculum, we changed the clicks on the angular adjustment apparatus; the number of clicks on the pure vertical adjustment apparatus was zero at all times (**Fig 1A**). The angular adjustment apparatus opens the blades prominently at their tips (apices or distalmost ends), but negligibly at their base.

The speculum was closed or mostly closed during sheath placement. After sheath placement, the speculum was opened to a predefined number of clicks: either five clicks for gloves or no sheath and three clicks for the condoms (the speculums allowed a maximum of seven clicks). We created this difference in opening protocol between the condoms and gloves because the gloves fit the speculum more tightly than the condoms and therefore the gloves restricted speculum opening by a greater amount. Therefore, the actual measured separation of the speculum blades for condoms versus gloves was more similar with three clicks for condoms and five for gloves. With that said, ultimately, our reported data for the vinyl gloves was for three clicks because some of the attempts to open the vinyl glove to four or five clicks cause the speculum to break (see **Results** section).

**Measurements and data collection.** After opening the speculum to the predefined number of clicks, we used a ruler to measure the speculum opening distance (initial height of speculum opening) for each trial. We defined this as the distance from the most superior part of the apex of the lower speculum blade to the most inferior part of the apex of the upper blade (**Fig 1A**). This measurement was done immediately after placement of the condom or glove as the opening distance was noted to gradually change over the following minutes as the materials deformed.

Then, an inflatable pressure cuff in the deflated state was circumferentially wrapped around the speculum blades so that it was flush with the speculum but not causing compression when

deflated. A high-resolution camera was mounted 10 cm behind the speculum, such that the camera was centered between and oriented parallel to the speculum blades (**S1 Fig**). For each trial, a photo was taken at baseline (0 mmHg); the cuff was then inflated to 200 mmHg (26.7 kPa, 272 cmH$_2$O) over a few seconds with photos captured at increments of 40 mmHg (5.3 kPa, 54 cmH$_2$O) for a total of 6 photos per trial. One author inflated the pressure while another triggered the camera remotely and a third monitored the quality of the data as it was collected. The same three authors (RSY, PKP, AH) performed the measurement and data collection for the entirety of the data.

The above sequence was performed for each trial for a total of 180 photos and 30 trials: 10 sets of 3 trials each under the same conditions (1 no sheath set as the control, 5 glove sheath sets, and 4 condom sheath sets). Three was chosen as the number of times a trial was repeated because it was a convenient sample size and there were prior studies to use for sample size estimation.

### Image labeling and processing

The compiled 180 images were saved as 2268px width by 4032px height colored JPEG files (average size 250 kilobytes) and uploaded to a free, online dataset labeling platform (Labelbox, San Francisco, CA, USA; April 2022; https://labelbox.com/). One co-author (RSY) acted as the labeler for all the images. The platform allowed the labeler to zoom and pan the image to precisely label the boundary.

The labeler was blinded to the trial information, including the pressure and experimental setup, except as was immediately obvious from viewing the image. The images were presented to the labeler in a semi-random fashion such that the pressures exerted (0–200 mmHg) were randomly distributed. The labeler examined all the images on the labeling platform and was tasked with drawing a rectangle on each image over the area that could be viewed through the speculum.

After labeling was completed, the labeled rectangle coordinates were downloaded. The horizontal distance in pixels between the labeled boundaries was calculated, representing the width of the field of view ("view width"), and used as the key metric to compare the different experimental conditions. This metric approximates the clinical field of view of the cervix. All of the horizontal distances were normalized by dividing by the horizontal distance of the baseline (0 mmHg) image within that respective trial and then subtracted from 100%. This yielded the percentage of the view (0–100%) obstructed by collapse of the simulated lateral vaginal walls. **Fig 2** shows labeled, representative images with the calculation of the view width. The full dataset of images, the subsequent calculations, interactive versions of the figures presented here, and all the necessary Python code to reproduce the calculations and figures have been uploaded to Zenodo repositories at https://doi.org/10.5281/zenodo.6790210 (dataset) and https://doi.org/10.5281/zenodo.6790202 (code).

### Statistical analyses

For each trial, there were a total of seven variables of interest: one ruler-based initial speculum opening distance measurement and six image-based calculations of the percentage of view obstruction (one calculation for 0 mmHg, 40 mmHg, 80 mmHg, 120 mmHg, 160 mmHg, and 200 mmHg). For each set of experimental conditions, we used the associated trials (N = 3) to calculate a mean and standard error of the mean (SEM) for each of the seven above variables at that experimental condition.

Repeatability of the model was measured with the Heise test-retest reliability coefficient [36], which can be conceptualized as a generalized version of applying Pearson's correlation

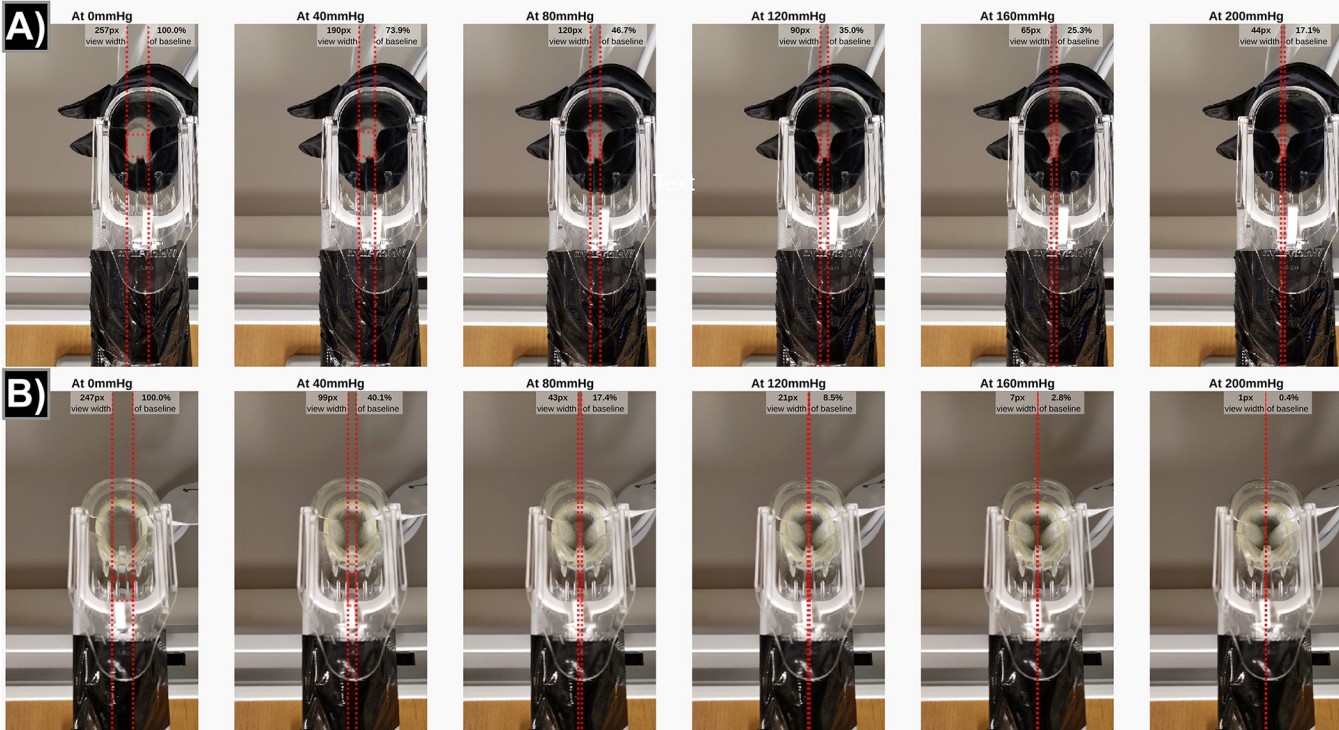

**Fig 2.** Sample of photos collected during a glove trial (a) and a condom trial (b). The resulting calculations for decreased visualization at each pressure are drawn on the figure. The calculation for relative width is indicated as the percentage of baseline and is equal to the view width at that pressure divided by the view width at 0 mmHg (baseline).

coefficient of the first trial measurements to the following (repeat) trial measurements. We also reported the mean of the Pearson's correlation coefficient between the first trial measurements and the second trial measurements, the second trial measurements and the third trial measurements, and the first trial measurements and the third trial measurements.

The mean and SEM were plotted to determine if any differences were clinically significant. For statistical significance, we performed two-tailed, unpaired, two-sample t-tests to compare the nitrile vs vinyl gloves and to compare the middle finger insertion technique vs the two-finger insertion techniques. The three different glove sizes were compared using a two-factor ANOVA test with pressure as the first factor and glove size as the second factor. Another two-factor ANOVA test, with pressure as the first factor and condom brand as the second factor, was used to compare the four different condom brands. Unless otherwise stated, a significance threshold of $p \leq 0.05$ was used.

## Results

### General findings

The physical model appears to have well-approximated the clinical scenario of lateral vaginal walls collapsing during a speculum exam. This is evidenced both subjectively from reviewing the photos and comparing them to clinical pelvic exams (**Fig 2**) and objectively because increasing the exerted pressure had the expected decrease in field of view due to the lateral wall collapsing (bowing) inward (**Figs 1B–1D**, **2**).

Additionally, the model was repeatable, indicated by low standard errors of the mean (SEM) using N = 3 trials per set. The means and SEMs of each set of trials are reported in

**Table 1. Composite table of the mean and standard error for all the sets of trials performed.**

| | | | | | Opening[c] Distance (mm) | | Calculated Percentage of Initial Visualization Obscured[d] | | | | | | | | | |
| | | | | | | | 40 mmHg (5.3 kPa, 54 cmH$_2$O[e]) | | 80 mmHg (10.7 kPa, 108 cmH$_2$O) | | 120 mmHg (16.0 kPa, 163 cmH$_2$O) | | 160 mmHg (21.3 kPa, 218 cmH$_2$O) | | 200 mmHg (26.7 kPa, 272 cmH$_2$O) | |
| Class | Material[a] | Size | Method | Clicks[b] | Mean | SEM | Mean | SEM | Mean | SEM | Mean | SEM | Mean | SEM | Mean | SEM |
|---|---|---|---|---|---|---|---|---|---|---|---|---|---|---|---|---|
| Glove | Vinyl | M | Middle finger | 3 | 11.2 | 0.93 | 6.5% | 3.2% | -6.5%[f] | 18.2% | 1.0% | 19.3% | 6.9% | 25.7% | 6.0% | 26.3% |
| | Nitrile | S | Middle finger | 5 | 27.3 | 0.17 | 33.4% | 8.6% | 53.4% | 8.0% | 70.9% | 10.5% | 78.8% | 9.5% | 85.1% | 8.4% |
| | | M | Two fingers | 5 | 51.0 | 0.58 | 49.8% | 4.1% | 73.8% | 4.1% | 87.2% | 4.7% | 93.9% | 2.9% | 98.2% | 1.2% |
| | | | Middle finger | 5 | 28.7 | 1.67 | 21.2% | 4.0% | 47.2% | 3.6% | 62.3% | 2.1% | 72.7% | 1.5% | 78.0% | 2.4% |
| | | L | Middle finger | 5 | 31.3 | 0.67 | 26.2% | 6.4% | 46.8% | 7.1% | 59.8% | 7.0% | 69.3% | 6.6% | 75.7% | 6.3% |
| Condom | Trojan (Latex) | | | 3 | 36.0 | 0.50 | 70.9% | 3.2% | 86.9% | 4.3% | 94.9% | 2.4% | 98.7% | 0.9% | 99.5% | 0.1% |
| | LifeStyles (Latex) | | | 3 | 36.3 | 0.17 | 65.6% | 4.7% | 88.4% | 5.2% | 93.9% | 2.8% | 98.3% | 0.7% | 99.6% | 0.0% |
| | Durex (Polyisoprene) | | | 3 | 35.7 | 0.17 | 58.6% | 1.4% | 81.1% | 3.1% | 90.9% | 3.4% | 96.9% | 1.1% | 98.8% | 0.5% |
| | Skyn (Polyisoprene) | | | 3 | 36.7 | 0.44 | 56.9% | 13.2% | 82.9% | 8.3% | 93.7% | 5.1% | 98.3% | 1.1% | 99.3% | 0.3% |
| None | | | | 3 | 41.0 | 0.00 | 66.1% | 8.9% | 84.4% | 5.8% | 91.9% | 3.9% | 95.2% | 2.2% | 98.4% | 0.7% |
| | | | | 5 | 54.0[g] | - | - | - | - | - | - | - | - | - | - | - |

S, small; M, medium; L, large; SEM, standard error of the mean (N = 3)

[a] Brand name substituted for material for condoms.

[b] Clicks indicates the amount of "clicks" of angular adjustment used on the speculum as shown in **Fig 1** (maximum is 7 clicks).

[c] Speculum opening distance was defined as the distance between the apices of the speculum blades as measured using a ruler immediately after placing the sheath and opening the speculum but before any external pressure was exerted.

[d] Calculated as the width of pixels still visible divided by the width visible at 0 mmHg.

[e] Vaginal pressure simulated around the speculum with values given in mmHg as well as the alternative units of kilopascal (kPa) and centimeters of water (cmH$_2$O).

[f] Negative value of small magnitude. This is because of negligible experimental error causing the recorded visualization at 80 mmHg (10.7 kPa, 108 cmH$_2$O) to be slightly better than the visualization at 0 mmHg, resulting in a negative value of the difference between them.

[g] Single value (N = 1) for speculum opening distance measured for 5 clicks of no sheath; full set of trials not planned in protocol

**Table 1**. The results were reproducible with a Heise test-retest reliability [36] of 0.887 for the width measurements and 0.970 for the speculum opening distance measurements (**S2 Table**). Alternatively, Pearson's correlation coefficients between the first and second, second and third, and first and third trial measurements were averaged to indicate reproducibility. These were 0.906 for relative width and 0.986 for speculum opening distance.

Finally, we found that some sheaths significantly restrict the opening of speculums (**Fig 3**). Thus, this unintended side effect needs to be weighed against any benefit from lateral wall retraction.

## Comparison of glove size and material types

The size of the gloves (small, medium, or large) had a statistically significant (p = 0.047), but clinically negligible, effect on the amount of lateral wall retraction (**Fig 4**). The small size vs large size groups had only a 6.6–11.1% difference in view obstruction (range depending on the amount of pressure exerted). Additionally, the large gloves did restrict the speculum opening the least amount; the difference was modest, but statistically significant (p = 0.011), amount (**Fig 3**).

Vinyl gloves were much tighter than their nitrile counterparts. The vinyl gloves were sufficiently stiff such that the plastic speculums repeatedly broke while attempting to open the speculum to the 5th click. We had to reduce the attempted speculum opening amount to three clicks to consistently prevent the vinyl gloves from breaking the speculums. While there was

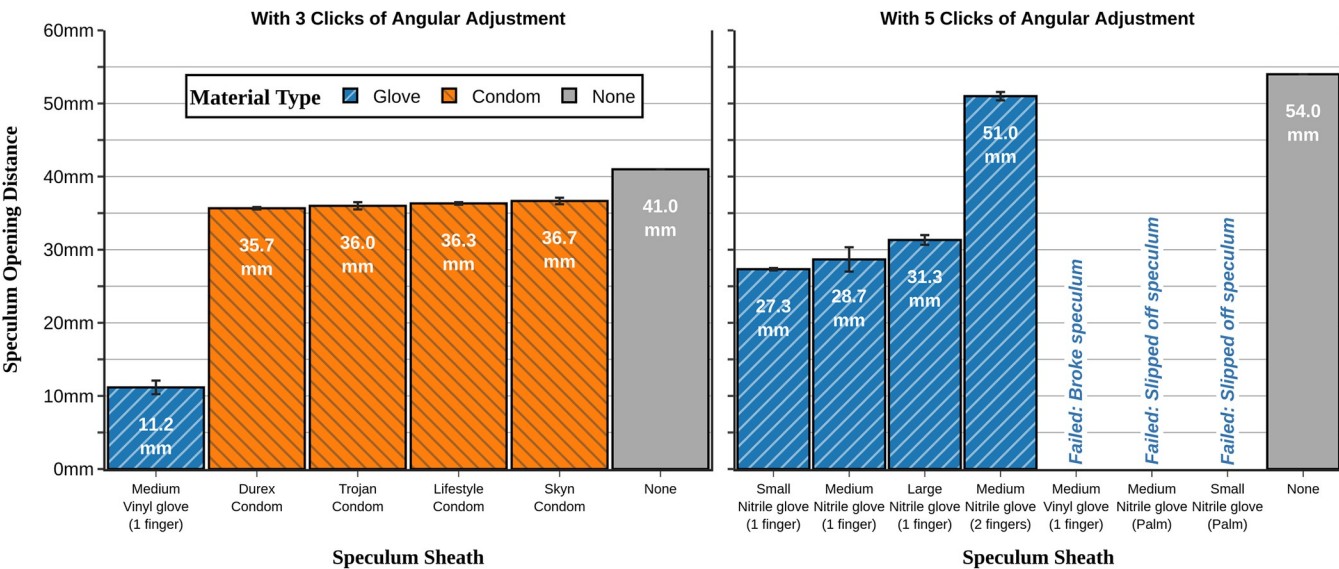

**Fig 3. Comparison of speculum opening distance by the type of sheath applied.** Restriction of this value is the disadvantage of using a sheath. Because the condoms were looser than the gloves, opening the speculums to the same number of clicks leads to widely different speculum opening distances (distances between speculum blade apices) depending on the sheath used. To keep the speculum opening distances comparable, the speculums were opened to 3 clicks for condoms (a) and 5 clicks for gloves (b). However, when the speculums were opened to 5 clicks for the vinyl material gloves, the speculums often broke. To give a numeric value, the vinyl material gloves are shown at 3 clicks as well (a), which was the maximum amount of clicks that could be consistently attained without speculum breaking. All measurements shown are at baseline, before any external pressure was exerted (0 mmHg). The speculums can be inserted into the gloves via one of three methods: "middle finger" (1 finger), "two fingers", and "palm" (see Fig 6). Trojan and LifeStyles condoms are composed of latex, and Durex and Skyn condoms are composed of synthetic polyisoprene.

minimal lateral collapse (**Fig 5**), the combination of the increased stiffness and the decreased number of click height led to a speculum opening distance of only 11.2±0.9 mm compared to 28.7±1.7 mm for the medium-sized nitrile gloves (**Fig 3**). This prevented visualization at any level of circumferential pressure using the vinyl gloves. Therefore, these trials may imply that nitrile gloves are a preferred material over vinyl gloves.

## Comparison among techniques of inserting the speculum into the glove sheath

In the absence of literature on the topic, we primarily used the middle finger of the glove as the sheath for the speculum. We also tested alternative methods such as (a) using two different

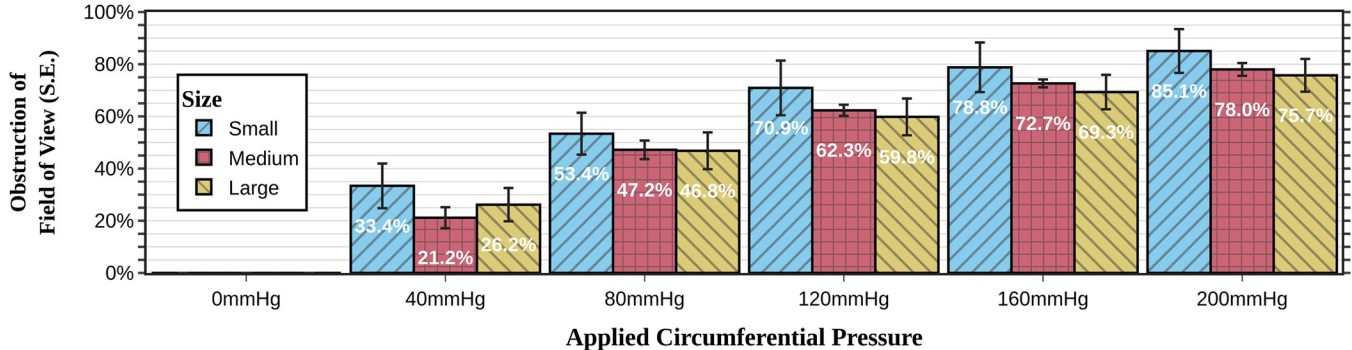

**Fig 4. Comparison of small, medium, and large glove sizes in preventing lateral vaginal wall collapse during speculum examinations.** The data shown is for nitrile gloves using the "middle finger" placement method.

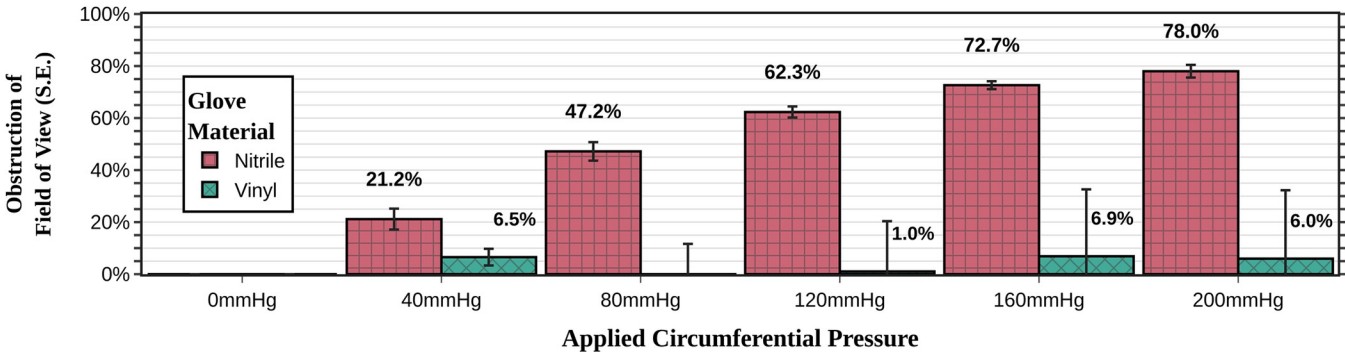

**Fig 5. Comparison of glove material type (nitrile vs vinyl) in preventing lateral vaginal wall collapse during speculum examinations.** Although the vinyl gloves seem better in this metric, they performed significantly worse overall because they restricted the speculum opening distance and occasionally even broke the speculum (see **Fig 3**). The data shown is for medium size gloves using the "middle finger" insertion method.

fingers for each bill, (b) using the entire palm. The two-finger method offered less restriction to speculum opening distance than the middle finger method (p<0.001). However, any benefit from this was greatly outweighed by its minimal lateral retraction power, which was only slightly better than no sheath (**Fig 6**). During the trial testing, the palm of the glove performed the worst with almost no lateral wall retraction. These trials were abandoned before completion when it became clear during experimental testing that this technique was impractical.

## Comparison of condom versus glove

The diameter of the condom is significantly greater than the diameter of any one of the individual fingers of a glove. Subsequently, condoms were fit more loosely on the speculum compared to the gloves and therefore restricted the speculum opening distance less. To provide a more reasonable comparison during our experiments, we opened the speculum to 3 clicks when the condom was applied versus 5 clicks for the glove (maximum of 7 clicks possible for the speculum). Both limited the opening distance to some degree; however, the condoms allowed for greater opening distance of the speculum (35.7–36.7 mm for condoms compared to 27.3–31.3 mm for gloves) (**Fig 3**). The condom provided minimal lateral wall retraction compared to having no sheath at all (**Fig 7**).

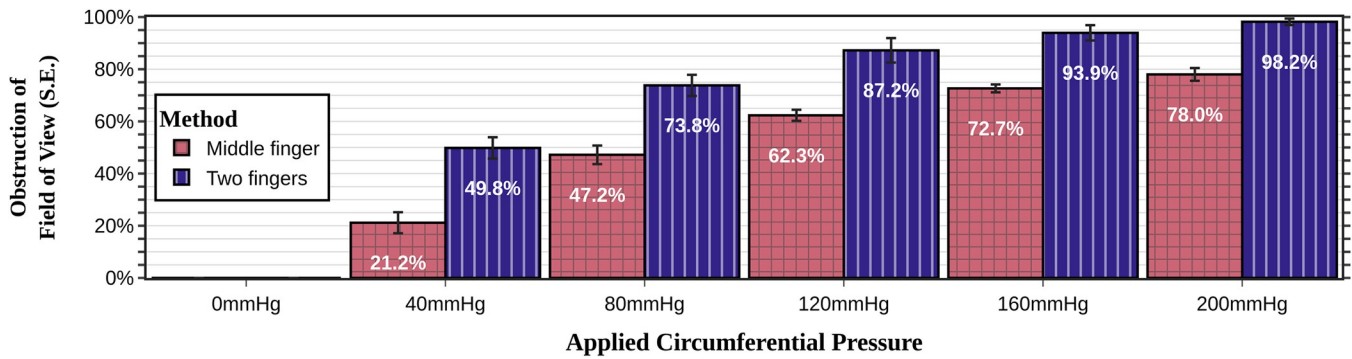

**Fig 6. Comparison of the different methods of applying the glove onto the speculum in preventing lateral vaginal wall collapse during speculum examinations.** The palm method (inserting the speculum into the palm of the glove only and using the palm as the sheath), is not shown as those trials were abandoned because of clear inferiority. (see **Fig 3**).

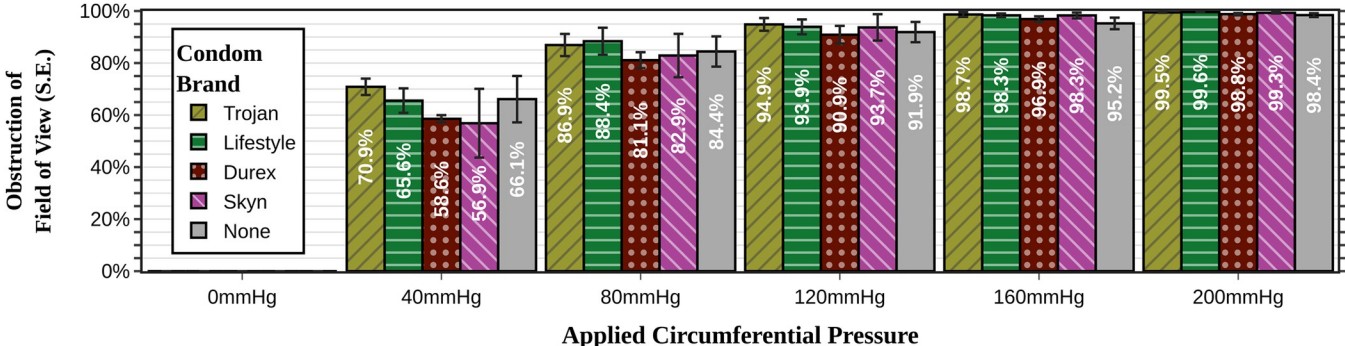

**Fig 7. Comparison of various condom brands and no condom in preventing lateral vaginal wall collapse during speculum examinations.**

## Comparison among the condom brands

The different condom brands performed similarly. There was no statistically significant difference in lateral wall retraction across pressures; they had similar initial speculum opening distances (**Fig 7**). However, the authors noticed many subjective differences during the trials. Trojan brand condoms frequently slipped back towards the base of the speculum and had to be readjusted. Additionally, different brands had different lubrication and different scents, which may impact either examiner or patient comfort.

## Discussion

Many practitioners turn to simple solutions they have at their disposal to improve access to the cervix during the office pelvic exam. This study aims to guide a practitioner's use of these materials with data from a simulation. It is a commonly held belief that using a condom or glove to sheath a speculum will improve visualization in patients with significant collapse of lateral vaginal walls, such as those who are pregnant or obese [7, 34]. An FDA approved speculum that has pre-built material attached to the sides of the speculum has been shown to improve visualization; however this tool is expensive and not widely used in a clinic setting [22]. Others are at even earlier stages of development [37–39]. Given this, we designed a study focusing on the evaluation of the most commonly used solutions in the clinic: the ad hoc solutions of condom and glove sheaths [7].

### Review of findings

**Table 2** lists our key findings and recommendations. Our study found that using a condom as a sheath for a speculum provide only a small benefit for decreasing lateral wall collapse. This represents a divergence from the commonly held intuition that using a condom does help with visualization in pelvic exams [7, 34]. Additionally, the different condom brands had minimal variation in their ability to affect visualization. However, only four condom brands of a similar size and two materials (latex, polyisoprene) were tested. Additionally, given that the condoms failed to provide retraction due to being too loose, the results may be different with varying speculum sizes.

Gloves greatly decreased lateral wall collapse; however, they restricted the opening of the speculum more than condoms. Size of the glove was not an important factor, but glove material was. Nitrile-material gloves allowed for greater speculum opening while still providing better lateral wall retraction compared to no sheath. Finally, inserting the speculum into only one finger of the glove was better than using two fingers or the palm of the glove. While these other

**Table 2. Summary of findings and recommendations.**

| Finding or Recommendation | Reference |
|---|---|
| Physical model to simulate the human vagina<br>A physical model composed of a speculum, a circumferential pressure apparatus, and a well-aligned camera can well represent cervical visualization during a pelvic exam, including lateral vaginal wall collapse. | **Figs 1B–1D, 2 and S2 Table** |
| Gloves vs condoms<br>Because gloves are tighter, they provide much better lateral wall retraction than condoms at the expense of some amount of speculum opening ability [a]. | **Figs 3, 4 and 7** |
| Nitrile vs vinyl gloves<br>Vinyl gloves greatly restrict speculum opening and thus impair visualization. The vinyl gloves may even break a plastic speculum. Therefore, they should be avoided in favor of nitrile gloves or condoms. | **Fig 3** |
| Glove placement method<br>If a glove is used, the speculum blades should be inserted into one of the glove fingers as opposed to multiple fingers or the palm of the glove. This maximizes lateral wall retraction. | **Fig 6** |
| Size and brand<br>The size of the glove or the brand or material of the condom have minimal impact on visualization. | **Figs 4 and 7** |
| Sheath hole<br>How well the hole is cut into the condom or glove can greatly affect the success of using the material as a sheath. | |

Key findings and recommendations from our experiments

[a] Nitrile gloves were overall better than condoms in our scenarios because the condoms fit loosely around the speculum. However, condoms may still be a reasonable option if the examiner is using an alternative speculum size than what was used in our experiments.

methods of speculum insertion into the glove were less restrictive of speculum opening, they did not provide appreciable lateral wall retraction.

## Relevance in the broader context: Obesity and other factors

Those who have increased abdominal mass, such as those who are pregnant or suffer from obesity or pelvic organ prolapse, are especially likely to have vaginal walls collapse inward and obscure the examiner's view. 89% of clinicians reported this as a major reason why cervical sampling is more difficult in patients with obesity [7]. The real-world consequences are significant. We have long known that obesity is linked to an increased risk of cervical cancer. However, a recent large study of epidemiologic data by Clarke et al demonstrated that obesity artificially "decreases" cervical precancer incidence through underdiagnosis, which consequently increases true cervical cancer incidence [9]. In other words, the connection between obesity and cervical cancer is caused by less successful screening in this population as opposed to the biological factors of obesity alone. Specifically, 20% of cervical cancers could be prevented if our tests for detecting precancer (secondary cancer prevention) reached a sensitivity that was the same in our patients who are overweight or obese as it is in those who are not [9]. This decreased sensitivity of detecting precancer is likely due to the decreased ability to visualize the cervix in this population leading to an inadequate tissue sample obtained during the Papanicolaou (Pap) smear [8, 9].

Improving cervical access in patients with obesity has the potential to improve more than the aforementioned 20% of cancer disparities. It is well-documented that obesity is an independent characteristic strongly linked with decreased adherence with cervical cancer screening recommendations [40–45], even when compared to other gynecologic cancers [41, 45, 46]. While the cause of the poor adherence is multifactorial, many of the causes could be improved by making the pelvic exam less difficult to perform. Physicians are known to be less willing to perform pelvic exams on patients who are obese [43, 46, 47]. Additionally, the struggle to perform good cervical exams increases the length of the exams and may subsequently increase

patient discomfort. Anticipated pain and the anxiety surrounding it are a primary reason why many women avoid pelvic exams.

While obesity is one important factor, the pressure that the lateral vaginal walls exert is multifactorial [48]. Whether or not the lateral walls will collapse on a speculum in any patient exam is affected by a multitude of factors, including menopausal status, use of estrogen, connective tissue diseases, and muscle tone [48, 49]. Approximations of vaginal wall mechanics have been described by researchers in the field of urogynecology. One estimate of the force exerted on the lateral vaginal walls ranged from 0.25–3 newtons [48]. Parameters of vaginal wall pressures have been found to change with age, but not as significantly with weight in a relatively small sample size [48]. Another study found that age and parity, not weight, were the primary attributes correlated with varying biomechanical pressure [49]. Weight, however, has been associated with increased risk of pelvic organ prolapse which reflects that increased weight leads to higher overall pressure on the pelvic organs [50]. While the role of obesity in the physiology of lateral vaginal wall collapse is not entirely understood, obesity remains an important clinical factor in a patient's gynecological health. Additionally, the obstruction of a physician's view during a pelvic exam, regardless of the underlying reason or the patient's body habitus, is certainly a medical problem that needs addressing.

### Limitations and areas for future work

There is a wide opportunity for future work and innovation in this area. This model was designed to simulate pressures exerted on the lateral walls of the speculum and can give results on how different materials work to counteract that. However, there was no testing performed on real patients and we cannot definitively exclude the existence of significant gaps between the physical model presented here and a clinical pelvic exam. While our model was internally valid meaning we could comfortably draw conclusions between different methods in the same circumstances, it can only approximate the actual pressures that a vagina exerts on a speculum.

Only four materials (nitrile and vinyl gloves; latex and synthetic polyisoprene condoms) and one speculum size were used throughout these experiments. Further evaluation could study different speculum sizes or investigate more condom types or glove materials (latex, polyethylene, polyurethane). Re-designing the speculum is also a prominent area of research [5, 12, 21–23, 31, 37–39, 51, 52] and future designs should consider the lateral walls exerting pressure on a speculum and limiting the visualization of the cervix during the pelvic exam.

Finally, patient comfort is notably missing from this experiment focused on a physical model. Variations in patient comfort during use of a condom compared to a glove would be important information for a practitioner to have in deciding how to proceed with an exam. Relatedly, the need for and effect of lubrication (either supplemental lubrication or the lubrication that already comes placed on the condoms) with these methods needs to be studied. In the case of Pap smears, this is especially important because some fear that lubricated speculums cause artifact on the samples making interpretation difficult [53–55]; However, many others disagree, believing this fear is unsubstantiated or that the effect is small compared to the benefit of pain reduction [4, 56, 57]. Either way, the results of prior studies might not hold in the case of lubricated condoms, especially as some have found the effect of lubrication of Pap smear quality to depend on the material properties of the lubrication [55].

### Summary of recommendations

If a practitioner is looking for a solution in their office to help with lateral wall collapse, these data suggest that they can use the middle finger of any size nitrile glove and proceed with the

exam. The main downsides of using the method are: (b) access to the lateral walls of the vagina is impaired and (b) this method slightly decreases the capability of the speculum to open completely. Condoms provide less wall retraction but may still be a reasonable option for alternate speculum sizes as they also affect speculum opening less. Vinyl gloves are an inadequate option and should be avoided. Additionally, regardless of the sheath used, the practitioner must take care that the sheath opening is cut wide enough so that the examiner does in fact get increased visualization while at the same time is not cut so large that the sheath slides backwards when the speculum is inserted into the vagina.

## Conclusion

We have presented a robust and simple physical model for simulating the collapse of the lateral vaginal walls during a pelvic exam and used it to compare inexpensive, ad hoc tools that clinicians often use practically. Through our simulations, we found that nitrile gloves are superior to vinyl gloves and to no sheath. In terms of lateral wall retraction, it also outperformed all of the four tested common condom brands. The size of the nitrile glove has little effect. Additionally, of the major methods to apply the glove onto the speculum, we demonstrated that placing the speculum blades inside the middle finger of the glove is superior to using multiple fingers or to using the palm. Using the latter two methods or using vinyl gloves is no better than or only minimally better than not using any sheath. While the nitrile glove is the best of the tested options, it still constricts the opening distance of the speculum to a certain degree.

These findings indicate that if a practitioner aims to decrease vaginal wall collapse with tools available in their office, they should use a nitrile glove of any size and place the speculum into only one of the fingers of the glove. This provides the most retraction of the lateral walls of the vagina with only some sacrifice of the opening ability of the speculum. This work aims to guide practitioners' use of materials frequently found in a clinical setting and used to prevent vaginal wall collapse during a speculum exam. Pelvic exams can be difficult to perform for patients whose lateral vaginal walls obscure the view of the cervix during the exam, which can lead to patient discomfort, wasted time, decreased adherence to screening recommendations, incomplete Pap smears, and ultimately worse cervical cancer outcomes. Improving visualization and ease of the exam for the practitioner directly improves the overall experience for the patient.

## Declarations

### Availability of data and materials

The compiled, machine-readable formatting of the dataset, the underlying captured photos, interactive versions of the figures presented here, the Python code and Jupyter notebooks that can reproduce all the analyses and figures, and all other files to support the findings of this study have been deposited in Zenodo, an online public repository, at https://doi.org/10.5281/zenodo.6790210 (dataset) and https://doi.org/10.5281/zenodo.6790202 (code).

## Supporting information

**S1 Fig. Photo of experimental setup.**
(PNG)

**S1 Table. Materials, software, and tools used in our experiments.**
(DOCX)

**S2 Table. Heise test-retest reliability and correlation comparing the measurements between the three trials of each condition combination.**
(DOCX)

**S1 Dataset. The dataset of measured speculum opening distances and image-derived lateral collapse.**
(XLSX)

## Author Contributions

**Conceptualization:** Rahul Sai Yerrabelli, Ashkhan Hojati.

**Data curation:** Rahul Sai Yerrabelli, Peggy K. Palsgaard, Ashkhan Hojati.

**Formal analysis:** Rahul Sai Yerrabelli.

**Investigation:** Rahul Sai Yerrabelli.

**Methodology:** Rahul Sai Yerrabelli.

**Project administration:** Amy J. Wagoner Johnson.

**Resources:** Rahul Sai Yerrabelli.

**Software:** Rahul Sai Yerrabelli.

**Supervision:** Rahul Sai Yerrabelli, Amy J. Wagoner Johnson.

**Validation:** Rahul Sai Yerrabelli.

**Visualization:** Rahul Sai Yerrabelli.

**Writing – original draft:** Rahul Sai Yerrabelli, Peggy K. Palsgaard.

**Writing – review & editing:** Rahul Sai Yerrabelli, Peggy K. Palsgaard, Ashkhan Hojati, Amy J. Wagoner Johnson.

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
