## [Decision Letter · Decision Letter 0]

1 Dec 2022

PONE-D-22-20240Improving visualization of the cervix during pelvic exams: A physical model and a steppingstone to reducing disparities in gynecological cancersPLOS ONE

Dear Dr. Yerrabelli,

Thank you for submitting your manuscript to PLOS ONE. After careful consideration, we feel that it has merit but does not fully meet PLOS ONE’s publication criteria as it currently stands. Therefore, we invite you to submit a revised version of the manuscript that addresses the points raised during the review process.

We look forward to receiving your revised manuscript.

Kind regards,

Salvatore Andrea Mastrolia, M.D.

Academic Editor

PLOS ONE

https://journals.plos.org/plosone/s/fileid=ba62/PLOSOne_formatting_sample_title_authors_affiliations.pdf.

Additional Editor Comments:

Dear Authors,

The Reviewer found your manuscript very interesting, moreover recommending a thorough revision in order to achieve publication.

I would suggest to take into consideration the Reviewer's comments, discuss and incorporate them within your manuscript in order to reach the standard requested for publication.

Best regards

Salvatore Andrea Mastrolia

PLOS One Academic Editor

Reviewers' comments:

Reviewer's Responses to Questions

**Comments to the Author**

1. Is the manuscript technically sound, and do the data support the conclusions?

Reviewer #1: Yes

2. Has the statistical analysis been performed appropriately and rigorously? 

Reviewer #1: Yes

3. Have the authors made all data underlying the findings in their manuscript fully available?

Reviewer #1: Yes

4. Is the manuscript presented in an intelligible fashion and written in standard English?

Reviewer #1: No

5. Review Comments to the Author

Reviewer #1: Title: Improving visualization of the cervix during pelvic exams: A physical model and a steppingstone to reducing disparities in gynecological cancers

Authors have conducted a clinical trial to determine the efficacy of many supportive methods during pelvic speculum examination that aims to visualize the cervix in a suitable way .An extremely beneficial way to overcome a clinical issue regarding the collapse of lateral vaginal wall. Many comments should be clarified before publication according to my best knowledge

1. The introduction is very long section, overcrowded, many paragraphs related to results , methods and discussion !!! please revise it as a whole

2. Page 16 , line 344-355 :The study is still a clinical trial conducted on human persons , it needs all the requirement during conducting the trial :1) Ethical approval to conduct it and to compare the results ; which is missing in the manuscript 2) sample size estimation !!! Missing from the manuscript!! You can write at the subsection that you have no references to use it for estimation and for that reason you used a convenient sample size!!! 3) How many women participated in the trial? For reach group? Does the same participant was used to test for the method of applying the speculum??? Or as you have mentioned in the methods section you have compare them with controls??? 4) How many control in relation to each case??? And which method you have used to include some of the participants as cases and others as controls!! A randomized or a non-randomized method 5)They are not controls rather being a comparable group 5)

3. Who did the examination?? Same person or a trained group? How did you have used the fit size of the speculum for each woman??

4. Where is the demographic data of the participant? Were they at reproductive age group? Paraous women or nulliparous? Any participant in menopausal age group?

Scientifically their vagina is different !!!

5. The setting of the study, time, design of the study should be recorded at the abstract and methods section of the text

6. The manuscript is overcrowded by comparing many methods and many sizes of the inspectors!!! It was much more suitable to compare 4 ways or methods with each group

7. Page 17, line 348: Capitations below the figures is specific for the title of the figure also may include any abbreviations have been used in it. Delete please the methods you have used from the legend of figure and replace it at the methods section

8. Page 17, Line 372: Trojan and LifeStyles condoms are composed of latex, and Durex and Skyn condoms are composed of synthetic polyisoprene: This information should not be recorded at the legend of the figure. Shift it to the methods section.

9. Figures (including photos of the speculums) and charts are not readable being of inadequate quality

6. PLOS authors have the option to publish the peer review history of their article (what does this mean?). If published, this will include your full peer review and any attached files.

Reviewer #1: **Yes: **SHAHLA KAREEM ALALAF

---

## [Author Response · Author response to Decision Letter 0]

17 Feb 2023

Please see the word document of the response to reviewer and editor comments for the best version. The text has been copied below.

We are grateful for all the feedback given on our manuscript. We have significantly revised the manuscript in the light of all the comments. The most notable revision was to make it clearer that our experiments were a simulation using a physical model and that no actual human subjects were involved. We made this clearer with multiple changes throughout the entire manuscript including the title, introduction, and methods, and discussion. We apologize greatly for any confusion before, and we hope you agree that the manuscript is improved with these changes. 

1. The introduction is very long section, overcrowded, many paragraphs related to results , methods and discussion !!! please revise it as a whole

We thank the reviewer for this feedback. In response, we have significantly shortened the introduction. In particular, the paragraph describing the speculum and lateral vaginal wall collapse (2nd paragraph of the original manuscript, lines 62-75) has been shortened as this is also described in the methods. Secondly, the two paragraphs focusing on what patient demographics this will affect (original manuscript lines 81-105) have been removed and merged into the discussion. Thirdly, significant portions of the final paragraph of the introduction (original manuscript lines 106-122) were removed or moved to the methods. Overall, the introduction was halved with a word counting decreasing from 1012 words to 560 words. We appreciate this feedback, and hope the introduction is clearer and more concise in the revision. 

2. Page 16 , line 344-355 :The study is still a clinical trial conducted on human persons , it needs all the requirement during conducting the trial :1) Ethical approval to conduct it and to compare the results ; which is missing in the manuscript 2) sample size estimation !!! Missing from the manuscript!! You can write at the subsection that you have no references to use it for estimation and for that reason you used a convenient sample size!!! 3) How many women participated in the trial? For reach group? Does the same participant was used to test for the method of applying the speculum??? Or as you have mentioned in the methods section you have compare them with controls??? 4) How many control in relation to each case??? And which method you have used to include some of the participants as cases and others as controls!! A randomized or a non-randomized method 5)They are not controls rather being a comparable group 5)

We thank the reviewer for their comment and are grateful for the opportunity to clarify the confusion. Our study was not conducted on human persons or animals. Instead, this study created a simulation of the compressive forces of the vaginal walls using mechanical equipment. The experiments were run on this simulation only. We have added additional clarifying statements in the introduction and in the discussion. Because no human persons or animals were involved as subjects in our experiments, no specific ethical approval was needed or sought. A sentence specifically stating this has been added to the first paragraph of the methods.

With regard to the n of trials (not the n of the number of people as there were none), there were 10 sets of 3 trials each. The details are given in the end of the “Measurements and Data Collection” of the methods. In response to the reviewer’s feedback, we added text to this section stating that there were no references for sample size estimation and thus we chose n=3 each as a convenient sample size. 

We hope that our revisions have made all of the above clearer. We are extremely grateful to the reviewer as we believe the manuscript is much easier to understand now that the revisions have been made.

3. Who did the examination?? Same person or a trained group? How did you have used the fit size of the speculum for each woman??

We thank the reviewer for their comment. We have added in information to the “Measurements and Data Collection” section of the methods describing who performed the measurements and data collection, and who performed the data analysis. The experimenter who performed the image labeling was blinded to the conditions of the experiment. The same speculum type (including size) was used for all cases (KleenSpec® Vaginal Specula, Item No. DYND70401L, Welch Allyn, Inc, UPC #732094143690); thus, we did not fit the size of the speculum to each specific case. As this was an initial first-of-its-kind study, we chose not to do this. However, we would recommend accounting for this in future studies as we suspect the differences would matter significantly. We included this in the limitations section. 

4. Where is the demographic data of the participant? Were they at reproductive age group? Paraous women or nulliparous? Any participant in menopausal age group?

Scientifically their vagina is different !!!

We agree that in the demographic information would be critical to a future study verifying our results in humans. However, our experiments were on physical stimulation model (please see response to comment #2 for our full explanation). We apologize for the confusion and hope our edits make this clearer. 

Although not exactly the information requested, we have added an Excel file of the dataset as supporting information. The entire code, images, dataset, and intermediate figures and calculations were uploaded to the Zenodo data repository.

5. The setting of the study, time, design of the study should be recorded at the abstract and methods section of the text

We thank the reviewer for this feedback. We have added a small paragraph to the beginning of the methods that hopefully better clarifies the type of study we are about to describe. We also took this information from the introduction (further addressing comment #1). We edited a sentence in the abstract to describe that these were controlled experiments with the physical model.

6. The manuscript is overcrowded by comparing many methods and many sizes of the inspectors!!! It was much more suitable to compare 4 ways or methods with each group

We thank the reviewer for their feedback and consideration. We agree this paper does compare multiple methods and has a significant number of variables. Because of this, we tried to summarize our results in a digestible and clinically relevant table (Table 2). We have also tried to significantly trim down the manuscript. Portions of the methods were moved to the supplement.

7. Page 17, line 348: Capitations below the figures is specific for the title of the figure also may include any abbreviations have been used in it. Delete please the methods you have used from the legend of figure and replace it at the methods section

We thank the reviewer for their comment. We removed this information from the legend of the figure 6 as recommended and changed the methods to reflect this. 

8. Page 17, Line 372: Trojan and LifeStyles condoms are composed of latex, and Durex and Skyn condoms are composed of synthetic polyisoprene: This information should not be recorded at the legend of the figure. Shift it to the methods section.

We thank the reviewer for their comment. We removed this information from the legend of the figure 7 as recommended. It has already been stated in the methods section so we did not need to move the text there.

9. Figures (including photos of the speculums) and charts are not readable being of inadequate quality

We thank the reviewer for their comment. It is not completely clear to us why the figures were not seen at high quality by the reviewer. We created a table (displayed below) of all the figure resolutions. As seen by the table, the figures were all of ≥300ppi and at least 2000 x 600px. We suspect that the journal decreased the resolution for the PDF proof version to decrease file space, but that the high-resolution version will be available to the reader in the published version. To view the high-resolution version now, the file can be downloaded instead of viewed through the pdf. Alternatively, we have included the pictures at the end of the response. We appreciate the reviewer’s time and hope any confusion was cleared up.

---

## [Decision Letter · Decision Letter 1]

3 Mar 2023

Improving visualization of the cervix during pelvic exams: A simulation using a physical model of the speculum and human vagina as a steppingstone to reducing disparities in gynecological cancers

PONE-D-22-20240R1

Dear Dr. Yerrabelli,

We’re pleased to inform you that your manuscript has been judged scientifically suitable for publication and will be formally accepted for publication once it meets all outstanding technical requirements.

Kind regards,

Salvatore Andrea Mastrolia, M.D.

Academic Editor

PLOS ONE

Additional Editor Comments (optional):

Reviewers' comments:

Reviewer's Responses to Questions

**Comments to the Author**

1. If the authors have adequately addressed your comments raised in a previous round of review and you feel that this manuscript is now acceptable for publication, you may indicate that here to bypass the “Comments to the Author” section, enter your conflict of interest statement in the “Confidential to Editor” section, and submit your "Accept" recommendation.

Reviewer #1: All comments have been addressed

2. Is the manuscript technically sound, and do the data support the conclusions?

Reviewer #1: Yes

3. Has the statistical analysis been performed appropriately and rigorously? 

Reviewer #1: Yes

4. Have the authors made all data underlying the findings in their manuscript fully available?

Reviewer #1: Yes

5. Is the manuscript presented in an intelligible fashion and written in standard English?

Reviewer #1: Yes

6. Review Comments to the Author

Reviewer #1: The corresponding author has responded to previous comments sufficiently. Technical improvement of pelvic examination is an important point to gynecologists for any indications it was.

Please remove from abstract the statement specifying the examination just to improve visualization of cervical malignancy

7. PLOS authors have the option to publish the peer review history of their article (what does this mean?). If published, this will include your full peer review and any attached files.

Reviewer #1: **Yes: **SHAHLA KAREEM ALALAF

---

## [Editor Report · Acceptance letter]

9 Mar 2023

PONE-D-22-20240R1 

Improving visualization of the cervix during pelvic exams: A simulation using a physical model of the speculum and human vagina as a steppingstone to reducing disparities in gynecological cancers 

Dear Dr. Yerrabelli:

I'm pleased to inform you that your manuscript has been deemed suitable for publication in PLOS ONE. Congratulations! Your manuscript is now with our production department. 

Kind regards, 

on behalf of

Dr. Salvatore Andrea Mastrolia 

Academic Editor

PLOS ONE